# Mitigating polyethylene-mediated periprosthetic tissue inflammation through MEDSAH-grafting

Jung-Wee Park[1]☯, Chong Bum Chang[1]☯, Young-Kyun Lee🅳[1], Jooyeon Suh[2,3], Jungsung Kim[4], Taejin Shin[5], YongHwa Kim[5], Donghyun Kang🅳[2,3‡]*, Jin-Hong Kim🅳[2,3‡]*

1 Department of Orthopedic Surgery, Seoul National University Bundang Hospital, Seongnam, South Korea, 2 Center for RNA Research, Institute for Basic Science, Seoul, South Korea, 3 Department of Biological Sciences, College of Natural Sciences, Seoul National University, Seoul, South Korea, 4 Corentec Co., Ltd., Seoul, South Korea, 5 R&D Center, Corentec Co., Ltd., Seoul, South Korea

☯ These authors contributed equally to this work.
‡ DK and JHK also contributed equally to this work.
* kangd@snu.ac.kr (DK); jinhkim@snu.ac.kr (JHK)

## Abstract

Periprosthetic tissue inflammation is a challenging complication arising in joint replacement surgeries, which is often caused by wear debris from polyethylene (PE) components. In this study, we examined the potential biological effects of grafting a [2-(methacryloyloxy)ethyl] dimethyl-(3-sulfopropyl)ammonium hydroxide (MEDSAH) polymer onto the surface of PE through a solvent-evaporation technique. J774A.1 macrophage-like cells and primary cultured mouse osteoblasts were treated with PE powder with or without the MEDSAH coating. MEDSAH grafting on PE substantially reduced the expression of pro-inflammatory cytokines and other mediators in primary cultured mouse osteoblasts, but did not significantly impact macrophage-mediated inflammation. Our findings suggest that a MEDSAH coating on PE-based materials has potential utility in mitigating periprosthetic tissue inflammation and osteolysis and preventing aseptic loosening in total joint replacements. Further research, including large-scale clinical trials and biomechanical analyses, is needed to assess the long-term performance and clinical implications of MEDSAH-coated PE-based materials in total joint arthroplasty.

## Introduction

Total joint replacement stands as a highly successful treatment option for osteoarthritis, rheumatoid arthritis, and other forms of arthritis impacting the major joints in the limbs [1]. Despite advancements in surgical methodologies and prosthetic implant design, wear particle-induced osteolysis around the prosthetic joints continues to be a troubling complication. This issue can exacerbate to non-infectious loosening of the artificial joints, which significantly limits their durability [2].

**Funding:** This work was supported by the Korea Medical Device Development Fund grant funded by Korea government (the Ministry of Science and ICT, the Ministry of Trade, Industry and Energy, the Ministry of Health and Welfare, and the Ministry of Food and Drug Safety; Project Number RS-2020-KD000038), grants from the National Research Foundation of Korea (NRF-2023R1A2C3003864 and NRF-2021R1I1A1A01055626), and the Korea Drug Development Fund funded by the Ministry of Science and ICT, the Ministry of Trade, Industry and Energy, and the Ministry of Health and Welfare (RS-2023-00217266). The funders had no role in study design, data collection and analysis, decision to publish, or preparation of the manuscript.

**Competing interests:** The authors have declared that no competing interests exist.

The development of periprosthetic osteolysis is recognized as an outcome of the inflammatory reaction of the host to wear debris generated from the bearing surface [3, 4]. Polyethylene (PE) particles are the most prevalent bone-resorptive particles within periprosthetic tissues, which originate from the interaction between the PE and metallic components in artificial joints [3, 5]. Macrophages recognize and phagocytose the PE particles, triggering the secretion of pro-inflammatory factors, including interleukin (IL)-1β, IL-6, prostaglandin-endoperoxide synthase 2 (PTGS2), and inducible nitric oxide synthase [6–9]. Along with this inflammatory response, these bone-resorptive factors further stimulate the upregulation of receptor activator of NF-κB ligand (RANKL, encoded by *TNFSF11*), a key molecule involved in osteoclastogenesis, ultimately leading to osteoclastic bone resorption [3, 10]. Therefore, limiting the generation of wear particles and mitigating the consequent inflammatory and bone-resorptive responses could potentially reduce the incidence of periprosthetic osteolysis.

In clinical settings, patients undergoing joint replacement surgery are at risk of revision surgery mainly due to loosening and wear [11]. Conventional ultra-high molecular weight polyethylene (UHMWPE) remains one of the most commonly utilized materials for liners or inserts in contemporary total hip and knee arthroplasties [12]. However, long-term observations of conventional PE liners in total hip arthroplasty [13–17] and total knee arthroplasty [18–20] indicated high rates of PE wear and wear-related revisions. Parilla et al. [17] recently identified a marked increase in wear-related revisions, manifested as severe liner wear, osteolysis, implant loosening, and secondary instability, along with a decline in survivorship between the 15-year (87.9%) and 25-year (61.1%) post-operative periods. These findings underscore the clinical significance of PE wear for both the short-term and long-term survivorship of implants.

A crucial approach to enhancing the durability and effectiveness of orthopedic implants involves the identification or development of innovative PE coating materials that can effectively reduce wear while minimizing the risk of unintended inflammatory reactions. [2-(Methacryloyloxy)ethyl]dimethyl-(3-sulfopropyl)ammonium hydroxide (MEDSAH) is a zwitterionic monomer comprised of molecular functional groups paired with quaternary ammonium cations and sulfonate anions (S1 Fig). This zwitterionic characteristic is recognized as conferring strong hydrophilicity to the surface, and a previous study demonstrated that zwitterionic polymer-modified surfaces had remarkably reduced wear due to the lubricating influence of the hydration layer [21]. Therefore, in this study, we developed a novel hip PE component grafted with MEDSAH and investigated its biological impact in terms of the inflammatory response and bone resorption effects *in vitro*.

## Materials and methods

### Production of MEDSAH-grafted PE particles

The MEDSAH polymer was grafted onto the surface of PE particles through a solvent-evaporation technique (S1 Fig). The raw UHMWPE particles were immersed in an acetone solution containing 10 mg/ml benzophenone for 30 s and then dried to remove the acetone at room temperature. The PE particles coated with benzophenone were then immersed in a 0.5 mM MEDSAH aqueous solution. Photo-induced polymerization was then carried out with ultraviolet (350 ± 50 nm) irradiation of 9 mW/cm$^2$ for 90 min at 60°C. Subsequently, the MEDSAH-grafted PE particles were washed with water and ethanol and then sterilized using gamma irradiation at 25 kGy.

### Tensile test

Tensile test specimens with a geometry of ASTM D638 Type IV were manufactured by Corentec Co., Ltd. Five materials were composed of UHMWPE and another five were composed of the MEDSAH-grafted UHMWPE compound. All test specimens were manufactured to a

thickness of 1.8 mm. The tensile tests were performed using a mechanical testing machine (ITEL-UTM101; SALT Co., Ltd.) with a crosshead displacement rate of 50.8 mm/min and the tensile strength at break was recorded for each specimen.

## Cell culture and PE powder treatments

The mouse macrophage-like cell line J774A.1 was purchased from the Korea Cell Line Research Foundation and maintained in RPMI-1640 medium supplemented with 10% fetal bovine serum (FBS), 100 U/ml penicillin, and 100 μg/ml streptomycin. The cells were treated with PE powder for various durations (24 hr, 48 hr, or 72 hr) and at different concentrations (50, 100, 500, 1000, and 5000 μg/ml) to assess time- and dose-dependent effects.

Mouse osteoblast isolation and primary culture were performed by using the method previously described by Suh et al. [22]. The animal experiments were reviewed and approved by the SNU Institutional Animal Care and Use Committee (IACUC no. SNU-210820-2). Briefly, mice were anesthetized by isoflurane and decapitated. Calvarial bone tissues were isolated from newborn ICR mice (day 0 to 3) and sequentially digested in 0.25% trypsin-EDTA for 20 min, followed by a solution containing 0.2% type II collagenase (Sigma Aldrich) for 30 min. The digestion solution was removed and the tissues were washed and immersed in a new digestion solution for 60 min. Finally, the cell suspension was filtered using a 70-μm cell strainer (VWR), centrifuged, and resuspended in growth medium comprising alpha-minimal essential medium supplemented with 10% FBS, 100 U/ml penicillin, and 100 μg/ml streptomycin. Experiments were conducted using primary cultured mouse osteoblasts that had been passed three to five times. The cells were treated with PE powder for various durations (24 hr or 48 hr) and at different concentrations (50, 100, 500, and 1000 μg/ml) to assess time- and dose-dependent effects.

Both J774A.1 cells and primary cultured osteoblasts were plated at a density of $3.0 \times 10^5$ cells per 35-mm cell culture dish and cultured under a 5% $CO_2$ atmosphere.

## RNA extraction, reverse transcription, and quantitative reverse transcription polymerase chain reaction (RT-qPCR)

Total RNA was extracted from the cells using TRI reagent (Molecular Research Center, Inc.) and reverse-transcribed by EasyScript Reverse Transcriptase (TransGen Biotech.). The obtained cDNA was then amplified by RT-qPCR using Power SYBR Green PCR Master Mix (ThermoFisher Scientific) on a StepOnePlus Real-Time PCR System (Applied Biosystems) with the following primers: *Ccl2* (forward, 5′-CAATGAGTAGGCTGGAGA-3′; reverse, 5′-TCTGGACCCATTCCTTC-3′), *Hprt* (forward, 5′-AGTCCCAGCGTCGTGATTAG-3′; reverse, 5′-GTATCCAACACTTCGAGAGGTC-3′), *Il11* (forward, 5′-AATTCCCAGCTGACGGAGATCACA-3′; reverse, 5′-TCTACTCGAAGCCTTGTCAGCACA-3′), *Il1b* (forward, 5′-CTTTGAAGAAGAGCCCATCC-3′; reverse, 5′-CATCTCGGAGCCTGTAGTGC-3′), *Il23* (forward, 5′-ATGCTGGATTGCAGAGCAGTA-3′; reverse, 5′-ACGGGGCACATTATTTTTAGTCT-3′), *Il6* (forward, 5′-GGAGACTTCACAGAGGATACCA-3′; reverse, 5′-TCATTTCCACGATTTCCCAGAG-3′), *Ptgs2* (forward, 5′-TGGGTGTGAAGGGAAATAAGG-3′; reverse, 5′-CTGCTGGTTTGGAATAGTTGC-3′), and *Tnfsf11* (forward, 5′-GTGAAGACACACTACCTGACT-3′; reverse, 5′- AACTTGGGATTTTGATGCTGG-3′).

## Results

### Pro-inflammatory effect of PE powder on macrophages and osteoblasts

Our study aimed to investigate the impact of PE powder with and without MEDSAH coating on the inflammatory responses of macrophages and primary cultured mouse osteoblasts. The

engulfment of PE wear particles by macrophages has been shown to trigger the release of various pro-inflammatory cytokines, including IL-1β, IL-6, and tumor necrosis factor (TNF) [23, 24]. Moreover, the challenge posed by PE wear particles was reported to interfere with the communication between osteoblasts and other cells within the microenvironment due to dysregulated production of cytokines [25, 26].

Prior to examining the impact of the MEDSAH coating on PE-induced inflammation, we first confirmed the capacity of PE powder to induce inflammation using the J774A.1 mouse macrophage-like cell line. In accordance with previous findings, the mRNA expression of the pro-inflammatory cytokine *Il6* was significantly upregulated 48 hr after treatment with PE powder in J774A.1 cells (S2 Fig). J774A.1 cells were then exposed to varying concentrations of PE powder (50, 100, 500, 1000, and 5000 μg/ml). It was observed that significant inflammatory responses began to manifest at a concentration of 500 μg/ml of PE powder (Fig 1 and S3 and S4 Figs). Similarly in primary cultured osteoblasts, the levels of inflammatory factors such as *Il1b* and *Il6* were significantly increased following treatment with PE powder (S5 Fig).

## Effect of MEDSAH grafting on PE

To explore the effects of MEDSAH grafting on PE, we first conducted mechanical testing to examine whether MEDSAH grafting significantly altered the mechanical strength of PE. The MEDSAH-grafted PE displayed a tensile strength of 40.50 ± 1.17 MPa, which did not differ significantly from that of the ungrafted PE of 42.89 ± 1.76 MPa (Fig 2). Consequently, we concluded that the grafting of MEDSAH onto PE does not alter its tensile strength, thus preserving its mechanical properties.

Next, we investigated the impact of MEDSAH grafting on modulating inflammatory responses in periprosthetic tissues. The MEDSAH-coated PE powder did not significantly affect the expression of *Il6*, *Ptgs2*, and *Il23* in J774A.1 macrophage-like cells when compared to the expression levels induced by PE powder without the MEDSAH coating (Fig 3). These results suggested that the MEDSAH coating does not have an apparent effect on modulating the PE-induced inflammation mediated by macrophages.

Meanwhile, MEDSAH-coated PE powder significantly reduced the expression levels of IL-6 family cytokines such as *Il6* (P < 0.0001) and *Il11* (P = 0.0108) in primary cultured mouse osteoblasts compared to those of the cells treated with the uncoated PE powder (Fig 4). The expression levels of other osteoblast inflammatory factors such as *Ptgs2* (P = 0.0297), *Tnfsf11* (P = 0.0023), and *Ccl2* (P = 0.0415) were also significantly reduced following treatment with MEDSAH-coated PE powder compared to those of mouse osteoblasts treated with the uncoated PE powder (Fig 5). These results thus indicated the potential utility of the MEDSAH coating on PE-based materials for mitigating inflammatory responses in osteoblasts within periprosthetic tissues.

## Discussion

Periprosthetic osteolysis is a clinical challenge in total joint arthroplasty, often leading to implant failure and revision surgeries [27]. Among the multiple factors implicated in the osteolysis process, PE wear particles have emerged as a key contributor. PE wear debris generated from the articulating surfaces of joint prostheses is known to elicit an inflammatory response around the peri-implant tissues [28, 29]. The particles can activate macrophages, resulting in the secretion of pro-inflammatory cytokines, chemokines, and growth factors [30]. These inflammatory mediators then promote osteoclast differentiation and activity [30] and inhibit osteoblast function [31], thereby leading to enhanced bone resorption and impaired bone

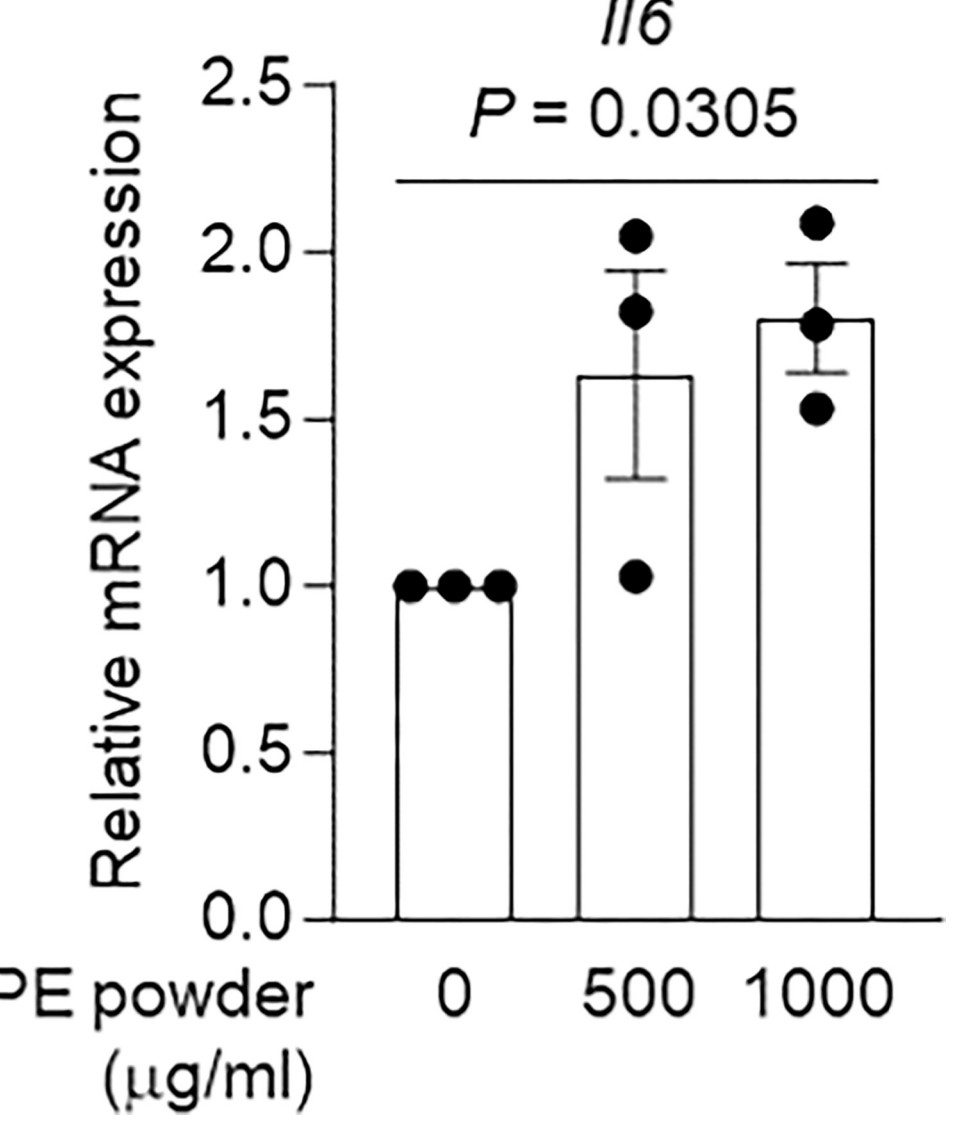

**Fig 1. Pro-inflammatory effect of PE powder on J774A.1 macrophages.** Relative mRNA expression of inflammatory factors according to various doses of treatment with PE powder in J774A.1 cells. Data indicate means ± standard error of the mean (SEM); significance was determined by one-way analysis of variance followed by Fisher's least-significant difference test.

formation. The net result is the progressive loss of periprosthetic bone and subsequent aseptic loosening of implant tissues.

Clinical studies have also demonstrated the impact of PE wear particles during the occurrence of periprosthetic osteolysis after total joint arthroplasty, including hip [29] and knee [18] arthroplasty. Analyses of periprosthetic tissues obtained during revision surgeries revealed the presence of wear debris in retrieved tissues, including the joint capsule tissues [32, 33] and synovial fluids [34, 35] from the hip and knee joints. Another study analyzed the peri-implant tissue from patients undergoing revision surgery and concluded that the concentration of PE wear particles that accumulated in the tissue was critically associated with osteolysis pathogenesis [36]. Moreover, the degree of wear particle accumulation has been associated to the severity of osteolysis [35]. Patients with a loose implant and subsequent osteolysis development

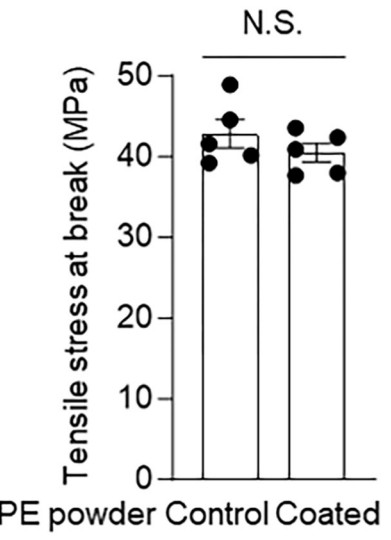

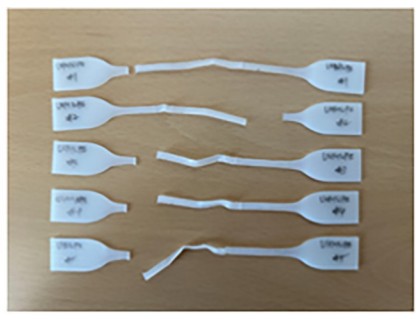

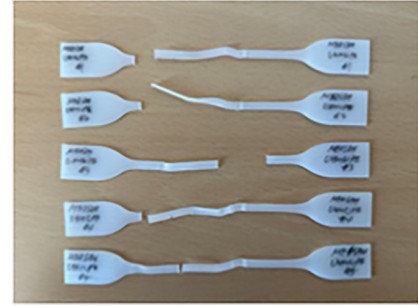

**Fig 2. Tensile strength of ungrafted and MEDSAH-grafted PE.** Tensile strength at break between the tensile test specimens composed of ungrafted PE and MEDSAH-coated PE (left). Tensile strength specimens after the operations on the mechanical testing machine (right). Data indicate means ± SEM; data were compared with two-tailed Student's t test.

showed an increased number of macrophages and higher expression of inflammatory cytokines [37, 38]. Notably, one study found a correlation in the levels of the inflammatory mediators interferon-gamma and toll-like receptor 4 in promoting aseptic loosening, while the level of the M2-polarizing cytokine IL-4 correlated with longer implant survival [39].

Given the significant influence of PE wear particles in the development of periprosthetic osteolysis, it is essential to consider approaches that can effectively alleviate these adverse effects. Anti-inflammatory therapies might attenuate the osteolytic processes, along with the use of wear-resistant materials, improved surgical fixation techniques, and the development of

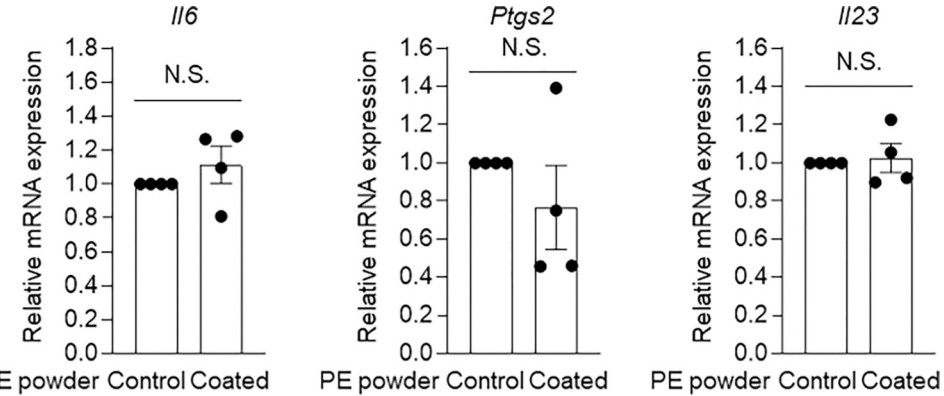

**Fig 3. Effect of MEDSAH grafting on PE particles in J774A.1 macrophages.** Relative mRNA expression levels of *Il6* and *Ptgs2* in J774.A1 macrophage-like cells treated with ungrafted and MEDSAH-coated PE powder (500 μg/ml). Data indicate means ± SEM; data were compared with two-tailed Student's t test.

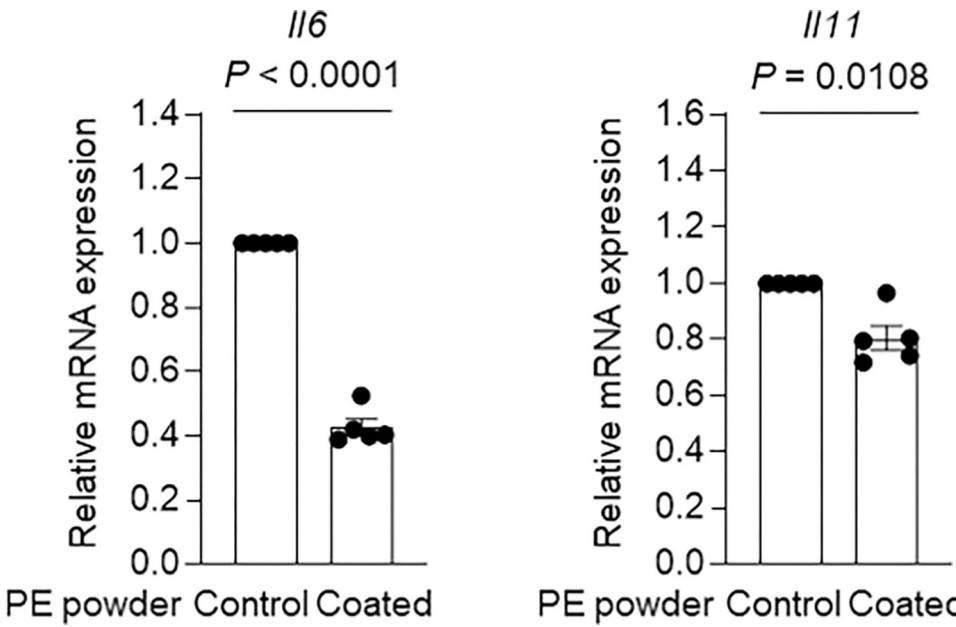

**Fig 4. Effect of MEDSAH grafting on the expression of IL-6 family cytokines in primary cultured osteoblasts.**
Relative mRNA expression of IL-6 family cytokines in primary cultured mouse osteoblasts treated with ungrafted PE powder and MEDSAH-coated PE powder (500 µg/ml). Data indicate means ± SEM; data were compared with two-tailed Student's t test.

an appropriate surface coating to reduce wear particle generation [40]. Our findings highlight the potential advantages of utilizing a MEDSAH coating on PE-based materials as a promising approach to mitigate inflammatory responses in osteoblasts within periprosthetic tissues. Compared to other strategies, application of a MEDSAH coating may offer a safer alternative as a bearing material, while effectively addressing the issue of osteolysis that frequently arises due to the loosening of total joint arthroplasty components. Based on these *in vitro* results, we anticipate that the implementation of a MEDSAH coating on PE-based materials will considerably enhance the efficacy of total joint arthroplasty by inhibiting periprosthetic osteolysis

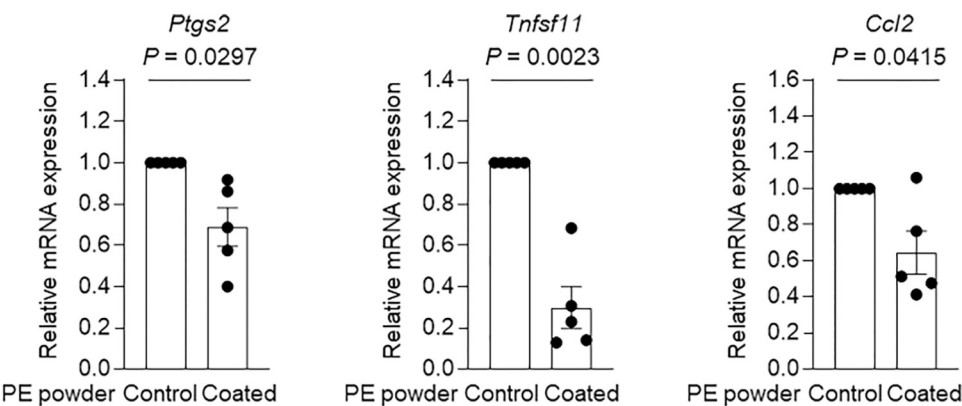

**Fig 5. Effect of MEDSAH grafting on the expression of other inflammatory factors in primary cultured osteoblasts.** Relative mRNA expression levels of inflammatory factors in primary cultured mouse osteoblasts treated with ungrafted PE powder and MEDSAH-coated PE powder. Data indicate means ± SEM; data were compared with two-tailed Student's t test.

and aseptic loosening, ultimately contributing to improved care and outcomes for patients undergoing total joint replacement procedures.

To further validate and expand upon these findings, it is crucial that future research efforts focus on conducting pre-clinical and clinical studies. Such investigations will not only aid in corroborating the present results but will also provide valuable insights into the long-term performance and clinical implications of MEDSAH-coated PE-based materials in total joint arthroplasty.

## Supporting information

**S1 Fig. Structural formula of [2-(methacryloyloxy)ethyl]dimethyl-(3-sulfopropyl)ammonium hydroxide (MEDSAH).**
(TIF)

**S2 Fig. Induction of inflammatory factors in response to treatment with 100 μg/ml ungrafted PE powder at varying time points in J774A.1 cells.**
(TIF)

**S3 Fig. Induction of inflammatory factors in response to treatment of varying doses of ungrafted PE powder after 24 hr in J774A.1 cells.**
(TIF)

**S4 Fig. Induction of inflammatory factors in response to varying doses of ungrafted PE powder after 48 hr in J774A.1 cells.**
(TIF)

**S5 Fig. Induction of inflammatory factors in response to various doses of PE powder doses for 24 hr in mouse osteoblasts.**
(TIF)

**S1 Table. Minimal data set.**
(XLSX)

**S1 Raw images.**
(PDF)

**S1 File.**
(DOCX)

## Acknowledgments

All individuals who contributed significantly to the work have been named as authors.

## Author Contributions

**Conceptualization:** Jung-Wee Park, Chong Bum Chang, Donghyun Kang, Jin-Hong Kim.

**Data curation:** Jung-Wee Park, Chong Bum Chang, Young-Kyun Lee, Jooyeon Suh, Jungsung Kim, Taejin Shin, YongHwa Kim, Donghyun Kang, Jin-Hong Kim.

**Formal analysis:** Jung-Wee Park, Chong Bum Chang, Young-Kyun Lee, Jooyeon Suh, Jungsung Kim, Taejin Shin, YongHwa Kim, Donghyun Kang, Jin-Hong Kim.

**Funding acquisition:** Young-Kyun Lee, Donghyun Kang, Jin-Hong Kim.

**Investigation:** Jung-Wee Park, Chong Bum Chang, Jooyeon Suh, Jungsung Kim, Taejin Shin, YongHwa Kim, Donghyun Kang, Jin-Hong Kim.

**Methodology:** Jung-Wee Park, Chong Bum Chang, Young-Kyun Lee, Donghyun Kang, Jin-Hong Kim.

**Project administration:** Jung-Wee Park, Chong Bum Chang, Young-Kyun Lee, Jin-Hong Kim.

**Resources:** Jungsung Kim, Taejin Shin, YongHwa Kim.

**Supervision:** Jung-Wee Park, Chong Bum Chang, Donghyun Kang, Jin-Hong Kim.

**Writing – original draft:** Jung-Wee Park, Chong Bum Chang, Young-Kyun Lee, Donghyun Kang, Jin-Hong Kim.

**Writing – review & editing:** Jung-Wee Park, Chong Bum Chang, Young-Kyun Lee, Jooyeon Suh, Jungsung Kim, Taejin Shin, YongHwa Kim, Donghyun Kang, Jin-Hong Kim.

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
