## [Decision Letter · Decision Letter 0]

22 Nov 2023

PONE-D-23-24916Mitigating polyethylene-mediated periprosthetic tissue inflammation through MEDSAH-graftingPLOS ONE

Dear Dr. Kim,

Thank you for submitting your manuscript to PLOS ONE. After careful consideration, we feel that it has merit but does not fully meet PLOS ONE’s publication criteria as it currently stands. Therefore, we invite you to submit a revised version of the manuscript that addresses the points raised during the review process.

Your manuscript has been evalauted by two experts in the filed and both of them suggested only min or revisions: please teke into consideration their suggestions.

We look forward to receiving your revised manuscript.

Kind regards,

Vittorio Sambri, M.D., Ph.D.

Academic Editor

PLOS ONE

Surface grafting of artificial joints with a biocompatible polymer for preventing periprosthetic osteolysis - https://doi.org/10.1038/nmat1233

In your revision ensure you cite all your sources (including your own works), and quote or rephrase any duplicated text outside the methods section. Further consideration is dependent on these concerns being addressed.

3. To comply with PLOS ONE submissions requirements, in your Methods section, please provide additional information regarding the experiments involving animals and ensure you have included details on (1) methods of sacrifice, (2) methods of anesthesia and/or analgesia, and (3) efforts to alleviate suffering.

“This work was supported by the Korea Medical Device Development Fund grant funded by Korea government (the Ministry of Science and ICT, the Ministry of Trade, Industry and Energy, the Ministry of Health and Welfare, and the Ministry of Food and Drug Safety; Project Number RS-2020-KD000038), grants from the National Research Foundation of Korea (NRF-2023R1A2C3003864, NRF-2016R1A5A1010764, NRF-2017M3A9D8064193, and NRF-2021R1I1A1A01055626), the Institute for Basic Science from the Ministry of Science, ICT and Future Planning of Korea (IBS-R008-D1), and Suh Kyungbae foundation (SUHF-18010068).”

“This work was supported by the Korea Medical Device Development Fund grant funded by Korea government (the Ministry of Science and ICT, the Ministry of Trade, Industry and Energy, the Ministry of Health and Welfare, and the Ministry of Food and Drug Safety; Project Number RS-2020-KD000038), grants from the National Research Foundation of Korea (NRF-2023R1A2C3003864, NRF-2016R1A5A1010764, NRF-2017M3A9D8064193, and NRF-2021R1I1A1A01055626), the Institute for Basic Science from the Ministry of Science, ICT and Future Planning of Korea (IBS-R008-D1), and Suh Kyungbae foundation (SUHF-18010068). The funders had no role in study design, data collection and analysis, decision to publish, or preparation of the manuscript.”

Reviewers' comments:

Reviewer's Responses to Questions

**Comments to the Author**

1. Is the manuscript technically sound, and do the data support the conclusions?

Reviewer #1: Partly

Reviewer #2: Yes

2. Has the statistical analysis been performed appropriately and rigorously? 

Reviewer #1: N/A

Reviewer #2: Yes

3. Have the authors made all data underlying the findings in their manuscript fully available?

Reviewer #1: Yes

Reviewer #2: Yes

4. Is the manuscript presented in an intelligible fashion and written in standard English?

Reviewer #1: No

Reviewer #2: Yes

5. Review Comments to the Author

Reviewer #1: The Authors aimed to examine the biological impacts of grafting [2-(methacryloyloxy)ethyl]dimethyl-(3-sulfopropyl)ammonium hydroxide (MEDSAH) to attenuate the inflammatory response induced by polyethylene wear particles.

The topic is interesting and the study is well designed.

Thew paper is well prepared. I have a few minor concerns on its regards.

Introduction should be limited to only relevant information. Also, I would detail further on the possible clinical effects of polyethylene wears in the clinical setting.

Has MEDSAH grafting on PE been studies on a biomechanics point of view?Does this modify the mechanical properties of polyethylene?

I would mitigate conclusions as these are not completely supported by results.

Many grammar and syntax errors. Please have the paper checked by a native speaker.

Reviewer #2: 98 … for 30 second —> for 30 seconds

110 … as previously described —> as described above

117 … primary cultured mouse osteoblasts passaged three to five times where used for experiments —> Experiments were conducted using primary cultured mouse osteoblasts that had been passed through three to five times.

6. PLOS authors have the option to publish the peer review history of their article (what does this mean?). If published, this will include your full peer review and any attached files.

Reviewer #1: No

Reviewer #2: No

---

## [Author Response · Author response to Decision Letter 0]

15 Jan 2024

Please see the uploaded "response to reviewers" document file.

---

## [Decision Letter · Decision Letter 1]

20 Mar 2024

Mitigating polyethylene-mediated periprosthetic tissue inflammation through MEDSAH-grafting

PONE-D-23-24916R1

Dear Dr. Kim,

We’re pleased to inform you that your manuscript has been judged scientifically suitable for publication and will be formally accepted for publication once it meets all outstanding technical requirements.

Kind regards,

Stuart Barry Goodman, MD PhD

Academic Editor

PLOS ONE

Additional Editor Comments (optional):

Reviewers' comments:

Reviewer's Responses to Questions

**Comments to the Author**

1. If the authors have adequately addressed your comments raised in a previous round of review and you feel that this manuscript is now acceptable for publication, you may indicate that here to bypass the “Comments to the Author” section, enter your conflict of interest statement in the “Confidential to Editor” section, and submit your "Accept" recommendation.

Reviewer #1: All comments have been addressed

2. Is the manuscript technically sound, and do the data support the conclusions?

Reviewer #1: Yes

3. Has the statistical analysis been performed appropriately and rigorously? 

Reviewer #1: Yes

4. Have the authors made all data underlying the findings in their manuscript fully available?

Reviewer #1: Yes

5. Is the manuscript presented in an intelligible fashion and written in standard English?

Reviewer #1: Yes

6. Review Comments to the Author

Reviewer #1: The Authors made good efforts in the attempt to ameliorate their paper. In my opinion, it is now suitable for t he publication on PlosOne

7. PLOS authors have the option to publish the peer review history of their article (what does this mean?). If published, this will include your full peer review and any attached files.

Reviewer #1: No

---

## [Editor Report · Acceptance letter]

14 May 2024

PONE-D-23-24916R1 

PLOS ONE

Dear Dr. Kim, 

I'm pleased to inform you that your manuscript has been deemed suitable for publication in PLOS ONE. Congratulations! Your manuscript is now being handed over to our production team.

Kind regards, 

on behalf of

Dr. Stuart Barry Goodman 

Academic Editor

PLOS ONE